# New Protocol for Production of Reduced-Gluten Wheat Bread and Pasta and Clinical Effect in Patients with Irritable Bowel Syndrome: A randomised, Double-Blind, Cross-Over Study

**DOI:** 10.3390/nu10121873

**Published:** 2018-12-02

**Authors:** Maria Calasso, Ruggiero Francavilla, Fernanda Cristofori, Maria De Angelis, Marco Gobbetti

**Affiliations:** 1Department of Soil, Plant and Food Sciences, University of Bari Aldo Moro, via Amendola 165/a, 70126 Bari, Italy; maria.deangelis@uniba.it; 2Interdisciplinary Department of Medicine–Pediatric Section, University of Bari Aldo Moro, Ospedale Pediatrico Giovanni XXIII, via Amendola 276, 70125 Bari, Italy; fernandacristofori@gmail.com; 3Faculty of Science and Technology, Free University of Bolzano, piazza Università, 5, 39100 Bolzano, Italy

**Keywords:** gluten, irritable bowel syndrome, lactic acid bacteria, wheat bread, pasta, diet

## Abstract

It has been suggested that sourdough fermented products have beneficial health effects. Fungal proteases and selected sourdough lactic acid bacteria were used to produce wheat bread and pasta with a reduced-gluten content (<50% of traditional products). Fermentable oligo-, di- and mono- saccharides and polyols and amylase/trypsin inhibitors were also evaluated. The sensorial features of new products were similar to traditional ones. The efficacy of these new products in reducing the severity of symptoms in Irritable Bowel Syndrome (IBS) patients were compared to traditional bread and pasta using a randomized, crossover-controlled trial. While on a strict gluten-free diet, patients were randomized to consume a reduced- or normal-gluten diet for 2weeks; then, patients from both arms started the wash-out period of one week, and subsequently started the final 2-week period on a normal or reduced-gluten diet. Compared to normal-gluten content, the administration of a reduced-gluten content diet resulted in a decrease of the Visual Analogue Scale score (*p* = 0.042), while no differences were found in the IBS-Severity Score, Hospital Anxiety and Depression Scale, and IBS-Quality of Life. Data herein reported are novel encouraging findings that should spur a new avenue of research aiming to develop products specifically designed for IBS patients.

## 1. Introduction

Irritable bowel syndrome (IBS) is one of the most common gastrointestinal (GI) disorders, with a prevalence of about 20% worldwide [1]. This high prevalence combined with the presence of persistent GI symptoms associated comorbidities and reduced quality of life [2] are responsible for relevant social costs and psychosocial burden [3]. Psychosocial stressors likely play an important role in IBS, and a recent meta-analysis reported an association between IBS and posttraumatic stress disorder [4].

According to ROME IV Criteria, IBS is a heterogeneous functional disorder commonly subtyped according to the prevailing bowel habit in IBS with constipation (IBS-C), diarrhea (IBS-D), and mixed with both constipation and diarrhea [5]. Although the etiology is still incompletely understood, it is considered a multifactorial disorder involving visceral hypersensitivity and hyperalgesia, abnormal motility, intestinal inflammation, and gut dysbiosis acting on a background of genetic susceptibility and psychiatric co-morbidity [6,7,8,9,10,11].

Numerous studies have demonstrated that particular foods elicit IBS symptoms (abdominal pain and bloating) in about 60% of patients [1], and accordingly, the exclusion of foods containing (i) fermentable oligo-, di- and mono- saccharides and polyols (FODMAPs); (ii) lactose, sucrose, fructose, or gluten-containing foods, and (iii) gas-producing foods and prebiotic dietary fibers, may result in clinical benefit [1,12,13]. Wheat contains several components including FODMAPs, gluten, amylase/trypsin inhibitors (ATIs), and wheat germ agglutinins, which are potentially able to trigger/worsen symptoms in IBS patients [14,15,16,17,18]. Based on theseobservations, several elimination diets have been attempted. The gluten-free diet (GFD) has been used toreduce IBS symptoms with conflicting results [14,19], considering that unless necessary, it should be prevented for nutritional, social, and economic reasons. However, there is an on-going debate about the role of wheat components on symptom induction in IBS patients [20,21]. Recent meta-analyses showed an improvement of symptoms in more than the 50% of IBS patients during low-FODMAPs diet [22,23]. There isno randomized controlled trial evidence to support the pathogenic involvement of gluten, ATIs, and wheat germ agglutinins in IBS [24]. Recently, a systematic review highlighted that there is insufficient evidence to recommend low FODMAP intake or GFD to reduce IBS symptoms [20,25].

In the last century, wheat breeding [26] and agricultural practices [27] have led to an abnormal increase in gluten content, while technological processing [28] and with very fast leavening times haveprevented gluten degradation prior to consumption. From a technological point of view, the reduction of gluten content may lead to the manufacture of wheat products with sensory and rheology properties not different from those of the high gluten counterparts [29].

The present study aims to define a protocol for manufacturing bread and pasta with reduced-gluten contents (<50% of standard products), and to assess theirefficacy and safety in patients with IBS using a randomized double-blind cross over study as compared to bread and pasta with normal-gluten content. Results of the present trial might improve our understanding of the role of innovative sourdough biotechnology in the production of fermented cereal-based foods, and help to identifyfuture directions for the development of products specifically designed for IBS patients.

## 2. Materials and Methods 

### 2.1. Sourdough Fermentation

Commercial *Triticum aestivum* and *Triticum durum* flours for bread and pasta, respectively, were kindly provided by MoliniTandoiSpA (Corato, Bari, Italy). The main characteristics of the *T. aestivum* and *T. durum* flours were the following: moisture, 15.5 and 15.6%; protein, 13.5 and 12.0% of dry matter (DM); fat, 1.0 and 0.7% of DM; ash, 0.5 and 0.9% of. DM; and total carbohydrates, 72.6 and 73.0% of DM, respectively. Gluten hydrolysis was performed as described by Rizzello et al. [29] with some modifications. Briefly, ten previously selected *Lactobacillus* strains (*L. sanfranciscensis* 7A, LS3, LS10, LS19, LS23, LS38 and LS47, *L. alimentarius* 15M, *L. brevis* 14G, and *L. hilgardii* 51B) with high peptidase activities were used. Selected fungal proteases (E1; 500,000 hemoglobin units on the tyrosine basis/g from *Aspergillus oryzae* and E2; 3000 spectrophotometric acid protease units/g from *Aspergillus niger*) (BIOCAT Inc., Troy, VA), which are routinely used as improvers in bakery industry, were also used. Gluten hydrolysis was carried out by sourdough fermentation using the following parameters: (i) each lactobacilli was used at final cell density in the dough of 8 log cfu/g; (ii) dough yield (dough weight 100/flour weight) of 200; (iii) fungal proteases E1 and E2 (ratio 1:1) 100 ppm. To achieve the 50% of gluten hydrolysis, the fermentation was performed at different time points (6, 8, 12, 18 and 24 h) at 30 °C under stirring conditions.

### 2.2. Microbiological and Immunological Analyses of Fermented Wheat Flours

Ten grams of dough fermented for 6, 8, 12, 18 or 24 h at 30 °C was homogenized with 90 mL of sterile saline solution. Lactic acid bacteria were counted by using modified MRS (mMRS, Oxoid, Basingstoke, Hampshire, UK), containing maltose 10 g/L, fresh yeast extract 50 mL/L (pH 5.6). Plates were incubated under anaerobiosis (AnaeroGen and AnaeroJar, Oxoid) at 30 °C for 48 h. The number of yeasts and molds were estimated on Sabouraud Dextrose Agar (SDA) (Oxoid) supplemented with chloramphenicol (0.1 g/L). Colonies were counted after incubation at 30 °C for 48 h. At each time point, the level of gluten was determined by using monoclonal R5 antibody in a specific enzyme-linked immunosorbent assay (ELISA). The analysis was performed using the Transia plate detection kit, following the instructions of the manufacturer (Diffchamb, Västra, Frölunda, Sweden) [30]. Sourdoughs fermented for 8 h a 30 °C, showing agluten reduction of 50%, were selected for further analyses. 

### 2.3. Wheat Bread and Pasta Making

After 8 h of fermentation, sourdoughs were freeze-dried to remove water. After milling, the resulting partially-hydrolyzed gluten flour was used to produce bread and pasta with reduced gluten content.

To produce bread with reduced gluten content, 3600 g of flour with partially-hydrolyzed gluten, 2160 mL tap water, and 2% (*w*/*w*) of baker’s yeast (corresponding to the final cell density of 8 log cfu/g) were mixed at 60 × *g* for 5 min, with a TVF50 plunging arm mixer (Domino SRL, Schio, Vicenza Italy) at the pilot plant of the MoliniTandoiSpA (Corato, Bari, Italy). Fermentation was at 30 °C for 1.5 h. Wheat flour sourdough bread, which contained non-hydrolyzed gluten, was manufactured according to the protocol routinely used in the company, and was used as the control (Appendix A). All types of breads weighed about 700 g. Fermentations were carried out in triplicate and bread was analyzed twice. Before baking, pH and total titratable acidity were measured. Total titratable acidity was determined on 10 g of samples, after grinding and homogenization with 90 mL of distilled water, and expressed as the amount (mL) of 0.1 N NaOH necessary to reach a pH of 8.3. The values of pH were determined by a Foodtrode electrode (Hamilton, Bonaduz, Switzerland). Bread was baked at 220 °C for 30 min (AGIV E310, AGIV, Verona, Italy).

Pasta, containing *T. durum* flour with partially hydrolyzed gluten was manufactured with the collaboration of the MoliniTandoi Spa personnel, using an SG30 pasta machine pilot plant (La Parmigiana s.r.l., Fidenza, Italy). Flour with partially-hydrolyzed gluten was mixed with different amounts of tap water (32, 35, and 37%) to find the optimal values of firmness and cooking loss. Ingredients were mixed for 5 min and each dough was left to hydrate for a further 5 min at 20 °C. The dough was then mixed for another 5 min and extruded through a teflon die n. 76 108 × 29 mm with 19 holes. The extruded material was cut with an electric rotating cutter to produce the pasta type “sedani”. The same type of pasta “sedani” containing non-hydrolyzed *T. aestivum* flour was manufactured according to the above protocol, and used as a control. For drying, pasta with both reduced content of gluten and normal content of gluten was arranged on frames (1.5 kg for frame) and treated in a ESS100 static dryer (La Parmigiana). Different drying cycles were tested to find the optimal number of cycles and temperature of drying. Finally, the normal (control) and reduced gluten content pasta samples were packed in plastic bags (Appendix A). 

### 2.4. Proximate Composition

Samples of bread and pasta were analyzed for energetic value, according to the method described in the Regulation of UE n.1169 of the 25/10/2011 using the converting factor Awater (4 kcal g^−1^ for carbohydrates and protein; 9 kcal g^−1^ for lipids). Moisture was determined according to the standard AACC method [31]. Fat, fatty acid methyl esters, trans, saturated, and unsaturated fatty acid levels were determined according to the standard AACC [31] and AOCS [32] methods. The total carbohydrate, crude protein, and crude fiber contents of the bread and pasta samples were determined using the method of AOAC [33]. Protein content was determined according to the ISO 1871:2009 method [34]. Total carbohydrates were calculated as the difference (100 − (N × 6.25 + lipids + ash + moisture). Mineral composition was determined according to the method n°. 33 reported in the Italian Ministry of Agriculture and Forests Ministerial Decree law of the 03 February 1989, for Na, Cl, K, and Ca and internal methods for Mg, P, Fe. The concentration of FODMAPs was determined using a Fructan HK enzymatic kit (Megazyme International Ireland Ltd., Bray, Ireland). 

Wheat ATIs from doughs, bread, and pasta samples were thoroughly extracted as reported by Zevallos et al. [35] using ammonium bicarbonate buffer, dialyzed, sterilized by filtration, and lyophilized. The lyophilized was dissolved in 10 mmol/L Tris-HCl, pH 8.0 and protein concentration was determined using Coomassie blue staining by Bradford method [36]. The albumin composition was analyzed by Tricine Sodium Dodecyl Sulfate Polyacrylamide Gel (SDS-PAGE) according to the procedure of Schägger and von Jagow [37] with some modifications [38]. The albumin ‘solutions were mixed with a sample buffer (0.05M Tris-HCl at pH 6.8, 4% (*w*/*v*) SDS, 12% (*v*/*v*) glycerol, 0.01% (*w*/*v*) bromophenol blue, and 2% (*v*/*v*) 2-ME) and heated for 3 min in a boiling water bath, then centrifuged at 10,000 rpm for 3min. Then 20μL of each sample was loaded into a well. Tricine SDS-PAGE was performed at 30V for 1h, and then up to 100V. After electrophoresis, the gels were stationed for 1 h in stationary liquid (45.4% (*v*/*v*) methanol, 9.2% (*v*/*v*) glacial acetic acid) and then stained for 12 h using 0.1% (*w*/*v*) Coomassie Brilliant Blue R-250. After staining, the gels were distained using a 50.0% (*v*/*v*) methanol, 10.0% (*v*/*v*) glacial acetic acid solution until a clear background was obtained. The protein standard was precision Plus Protein^TM^ Standard Plugs unstained (BioRad Laboratories srl, Segrate, Milan, Italy). The monomer, dimer, and tetramer forms of ATI were recognized according to their molecular weights, and have been grouped into three subfamilies of approximately 50–60, 24–30, and 12–15 kDa, based on their electrophoretic mobility, respectively [38,39].

### 2.5. Texture and Image Analysis

Instrumental texture profile analysis was carried out with a TVT-300XP Texture Analyzer (TexVol Instruments, Viken, Sweden) equipped with a cylinder probes P-Cy25S. For bread, boule-shaped loaves (700 g) were stored at room temperature for 24 h after baking. For the pasta analysis, samples were cooked forthe optimal cooking time, left to cool at room temperature, and placed in a beaker (diameter, 100 mm; height 90 mm), filled to about half volume. The analysis was carried out by applying two compression cycles at a speed of 1 mm/s and 30% deformation of the sample [40]. The results of the texture profile analysis were acquired and elaborated using the Texture Analyzer TVT-XP 3.8.0.5 software (TexVol Instruments, Viken, Sweden), giving the following bread textural parameters: hardness (maximum peak force), fracturability (the first significant peak force during the probe compression of the bread, and resilience (ratio of the first decompression area to the first compression area). For pasta, there wereprimary parameters (hardness, cohesiveness, and springiness) and secondary parameters (resilience, gumminess and chewiness) [40]. The specific volume of the breads was measured by the BVM-test system (TexVol Instruments). 

### 2.6. Color Measurement

Color was measured in three different points of bread crust or pasta samples using a Minolta CR-10 Camera [29,40]. The CIELAB L, a, bcolor space analysis method was used, where L represents lightness (white-black) and a and b the chromaticity co-ordinates (red-green and yellow-blue, respectively). Results were reported in the form of a color difference, dE* ab, as follows: dE*ab = √((dL)^2 + (da)^2 + (db)^2).

Where dL, da and db are the differences for L, a, and b values between sample and reference (a white ceramic plate having L = 93.4, a = −1.8, and b = 4.4).

### 2.7. Hydration Test

Five grams of each pasta sample were placed in a beaker containing 100 mL of tap water (ratio pasta: water of 1:20), which was placed in a thermostatic bath at 25 °C. After 5, 10, 15, 30, 60, and 180 min incubation, samples were removed from water, drained for 1 min, carefully blotted with tissue paper to remove superficial water, and weighed. The results were expressed as ((W1-W0)/W0) *100, where W1 is the weight of the hydrated sample and W0 is the weight of the dry sample [40].

### 2.8. Cooking Loss and Water Absorption

The method of Schoenlechner et al. [41] was used to determine the cooking time. Twenty-five grams of pasta were put into a beaker containing 300 mL of boiling water (without salt addition). Every minute, some pasta pieces were taken out and pressed between two Perspex plates. The optimal cooking time corresponded to the disappearance of the white core.

Cooking loss was evaluated by determining the amount of solid losses into the cooking water [42]. Portions of 30 g of pasta were cooked in 300 mL of boiling tap water (ratio pasta: water of 1:10) without salt addition. Pasta samples were cooked for the optimal cooking time. After cooking, the volume of water was brought to the initial volume. Dry matter was determined on 25 mL of freeze-dried cooking water. The residue was weighed, reported as percentage of the dry material, and expressed as grams of matter loss/100 g of pasta. The increase of pasta weight during cooking was evaluated by weighing pasta before and after cooking. The results were expressed as ((W1-W0)/W0) *100, where W1 is the weight of cooked pasta and W0 is the weight of the uncooked sample.

### 2.9. Sensory Analysis

The sensory analysis of the breads and pasta samples was carried out by 10 panelists (5 males and 5 females, mean age: 35 years, range: 18–54 years) according to the method described by Rizzello et al. [29] and Curiel et al. [40]. The sensory attributes for bread and pasta analysis were discussed with the panelists during the introductory sensory training sessions. All procedures (test conditions, preliminary discussion and test, selection, training and monitoring of the assessors, choice of the descriptors and appropriate scale, and evaluation of results) were carried out as suggested by the guidelines n.13299 of the International Organization for Standardization [43]. Four hours after baking, global acceptance, cereal-like odor, yeast odor, cooked odor, cereal-like taste, fermented taste, acid, sweetness, salty, bitter, staling, elasticity, moisture, crumb color, and crust color were considered as sensory attributes of the breads. 

For the pasta analysis, the sensory attributes—global acceptance, cereal-like odor, fermented odor, corn-like flavor, cereal-like taste, fermented taste, corn-like taste, acid, sweetness, salty, cooking homogeneity, stickiness, texture, and gumminess—were considered. A scale from 0 to10, with 10 being the highest score, was used. Samples were served in random order and evaluated in two replicates by all panelists.

### 2.10. Patients and Study Design

A total of 100 consecutive patients referred for IBS from January 2014 to December 2016 from Apulia region were invited to participate in the study. Inclusion criteria were: (1) age above 18 years; (2) having bowel symptoms (altered bowel habit, lower abdominal pain, bloating, or distention suggestive of IBS), as determined by Rome III criteria [44]. Exclusion criteria were: (1) celiac disease (CD) according ESPGHAN (European Society of Pediatric Gastroenterology Hepatology and Nutrition) guidelines and wheat allergy (Skin prick test and/or wheat and gluten specific IgE); (2) previous GI malignancy and/or surgery; (3) cardiovascular, hepatic, respiratory, endocrine, renal, hematologic, neurologic, or psychiatric disorder; (4) pregnancy or lactation; (5) alcohol abuse and/or drug addiction; (6) corticosteroids or anti-inflammatory drugs use in the last three months, and (7) participation in another clinical trial within 6 months before enrolment. The trial included three phases: the first phase was aimed to monitor the severity of IBS symptoms while on habitual diet, the second while on a run-in open gluten-free diet, and the third phase during the double blind cross over challenge. Throughout the trial, an experienced dietitian instructed all patients in order to guarantee a well-balanced diet with proper gluten avoidance without modification to dietary habits. The design of the clinical study is reported in Figure 1. 

The study was carried out and the data were collected at outpatient centers of the Interdisciplinary Department of Medicine, University of Bari Aldo Moro.

### 2.11. Randomization and Blinding

The randomization was performed by an independent researcher using a computer-generated randomizationlist in blocks of 6 patients (i.e., 3 receiving reduced-gluten diet and 3 normal-gluten diet in random order within the block). Labeling of study products wasperformed by an outside partner not taking part in the study, so that pasta and bread contained reduced- or normal-gluten contents were indistinguishable in appearance. The success of blinding was verified during two different panel tests. Despite some differences in sensory characteristics (see paragraph 3.5), 10 out of 10 participants at the panel test were unable to distinguish which bread and pasta was the low gluten content one.

### 2.12. Double Blind Cross Over Challenge

While on a strict GFD, patients were randomized to consume a reduced- or normal-gluten diet for 2-weeks; then, patients from both arms started the wash-out period of one week, and subsequently started the final 2-week period on normal or reduced-gluten diet.

We defined a diet with a reduced content of gluten as a GFDplus 100 g of reduced-gluten pasta and 200 g of reduced-gluten bread per day, and a diet with a normal content of gluten as a GFDplus 100 g of normal-gluten pasta and 200 g of normal-gluten bread per day. Representative sachets of pasta and bread contained reduced- or normal-gluten content were marked with a serial number (Appendix A).

In order to assess compliance to study treatment and to record adverse events, patients were interviewed on a regular basis by medical personnel blinded to the regimen; compliance was calculated as the percentage of returned study product,and compliance was considered acceptable if >80%.

The study adhered to the Declaration of Helsinki and was approved by the Ethical Institutional Review Board of Institutional Ethics Committee of Bari University Hospital (DG no. 703/2013). Written informed consent was obtained from the patients, who were fully informed of the nature and purpose of the study. The study was registered in the Protocol Registration System Clinical Trial.gov(ClinicalTrials.gov Identifier: NCT03638544).

### 2.13. Outcomes 

Primary outcome was IBS severity measured by Irritable Bowel Syndrome Severity Score (IBS-SS) and Visual Analogue Scale (VAS). IBS-SS consists of five questions that generate a maximum score of 100 each using a visual analogue scales and leading to a total possible score of 500; mild, moderate, and severe cases were indicated by scores of 75 to 175, 175 to 300 and >300, respectively [45]. VAS is a scale with severity descriptors (absent, very mild, mild, moderate, severe, very severe) where 0 is no pain and 10 the worst possible pain [46]. Secondary outcomes were: (i) health related quality of life measured by Hospital Anxiety and Depression Scale (HADS) and (ii) Irritable Bowel Syndrome Quality of Life (IBS-QoL) [47]. A symptomatic response characterized by a decrease of at least 30% of the global VAS (or IBS-SS) from the normal-gluten content and reduced-gluten content diet was defined as a clinical response [48].

### 2.14. Statistical Analysis

The sample size was calculated assuming a 35% difference in response between reduced-gluten and normal-gluten products. We estimated that 19 patients would be required for the study to have 80% power and an α error of 5%. Fermentations were carried out in triplicate and each analysis was repeated twice. All the data were subjected to one-way analysis of variance (ANOVA), and pair comparison of treatment means was achieved by Tukey's procedure at a *p* value of 0.05. Data obtained from an in vivo gluten challenge were subjected to permutation analysis using PermutMatrix [49] and Principal Components Analysis (PCA) using Statistica 7.0 for Windows. A per-protocol analysis was applied to the trial. Normally-distributed grouped data were expressed as the means and standard deviation (±SD) and compared using paired and unpaired t-tests. Non-parametric grouped data were expressed as means (95%CI) and compared using the Mann-Whitney rank sum test (paired) or Wilcoxon´s signed rank test (unpaired). The randomization list was generated using the online resource available at www.randomization.com. Proportionate data were compared using Fisher´s exact test or the χ2 test as appropriate. Differences between groups were analyzed using the two-tailed Student t test for independent samples. A *p* values <0.05 were regarded as significant. The statistical software Statistica for Windows (Statistica 7.0, Milan, Italy) was used.

## 3. Results

### 3.1. Characterization of Sourdoughs

Sourdough fermentation with ten *Lactobacillus* strains and fungal proteases was applied to reduce gluten content of *T. durum* and *T. aestivum* wheat flours of about 50%. As determined by plating on modified MRS, the cell density of lactic acid bacteria ranged from about 9.01 (sourdough fermented for 6 h) to 9.3 (sourdoughs fermented for at least 8 h) log cfu/g. The cell density of yeasts ranged from 1.5 to 3.8 log cfu/g in sourdoughs fermented for 6 and 24 h, respectively. Molds were not found in any sourdoughs. No statistical differences (*p* > 0.05) were found between viable counts of lactobacilli and yeasts in *T. durum* and *T. aestivum* wheat sourdoughs (data not shown). 

As determined by R5-ELISA, the untreated *T. durum* and *T. aestivum* wheat flours used for sourdough fermentation contained ca. 75,000 ± 1650 and 74,060 ± 814 ppm of immune reactive gluten, respectively. Under our experimental conditions, there was a degradation of about 50% of native immune reactive gluten after 8h of fermentation (38,650 ± 810 and 35,960 ± 1550 ppm of immune reactive gluten in *T. aestivum* and *T. durum* wheat hydrolyzed flour, respectively).

Before freeze-drying, the selected *T. aestivum* sourdoughs containing 50% reduced gluten had values of pH and total titratable acidity of 4.35 and 6.3 mL of 0.1 N NaOH, respectively. No statistical differences (*p* > 0.05) were found between *T. durum* and *T. aestivum* wheat sourdoughs (data not shown).

### 3.2. Chemical and Nutritional Characteristics of Wheat Bread and Pasta with Reduced Content of Gluten

At the end of 8 h of fermentation, wheat hydrolyzed flours were subjected to spray drying, and used for the production of bread and pasta. Bread was produced using *T. aestivum* hydrolyzed flour, tap water, and baker’s yeast without the addition of structuring agents. This bread was compared to traditional sourdough bread made with not hydrolyzed flour. Pasta was produced using *T. durum* hydrolyzed flour and water without the addition of structuring agents. Optimal dough moisture (37%) was chosen through the evaluation of firmness and cooking loss during several trials (data not shown). Based on firmness and cooking loss, the optimal dynamic drying cycle was selected (see Appendix A). Pasta made by traditional protocols (*T. durum* not hydrolyzed flour) was used as control.

Breads with normal or reduced gluten content were not different in terms of moisture, fat, fatty acid methyl esters, saturated fatty acids, monounsaturated fatty acids, polyunsaturated fatty acids, total carbohydrate, protein, crude fiber, and minerals (Na, Cl, K, Ca, Mg, P and Fe) content (Table 1).

We found no differences for the chemical composition of pasta containing normal or reduced gluten content with the soleexception of Ca, whose level was lowest in reduced-gluten pasta (Table 1). Compared to the control, bread with reduced gluten content showed the same level (*p* = 0.182) of FODMAPs content (about 0.9%). Non-significant differences (*p* = 0.099) were observed for FODMAPs between pasta samples. Compared to flour, the concentration of ATIs decreased from ca. 2.3 to 1.7 mg/g during the 8 h of fermentation at 30 °C. No statistical differences were found between fermented doughs. The intensity of the tetrameric, dimeric, and monomeric ATI bands was similar between the normal-gluten and reduced-gluten bread and pasta (Appendix A).

### 3.3. Texture and Color Characteristics

The treatment of wheat flour with lactobacilli and fungal proteases only slightly affected the structural, image, and color features of the resulting bread and pasta (Table 2).

Compared to bread with normal-gluten content, the specific volume of the bread made with hydrolyzed flour slightly (*p* = 0.047) decreased. As shown by the textural profile analysis, the hardness was similar (*p* = 0.291) between breads with normal and reduced gluten content. In contrast, the fracturability point, corresponding to the force at the first significant break during compression of the bread, was lowest for the bread with reduced gluten content. The same was found for the resilience (how well the bread fights to regain its original position). The crumb grain of the two breads was evaluated by image analysis technology. Digital images were pre-processed to estimate crumb cell-total area through a binary conversion. Compared to normal-gluten bread, the cell-total area (corresponding to the black pixel total area) of the reduced-gluten bread was only slightly lower, and showed the lowest crust lightness (L) and the highest value of dE*ab.

The use of reduced-gluten flour only slightly affected the texture profile analysis parameters of the pasta: it showed a slightly lower value of hardness (*p* = 0.034) compared to the normal-gluten pasta. No significant differences were found for cohesiveness, resilience, springiness, gumminess, and chewiness. For the color characteristics, pasta with reduced gluten content showed the highest dE*ab values.

### 3.4. Hydration Test, Cooking Loss and Water Absorption

Water absorption capacity was investigated with the aim of evaluating how ingredients and processing conditions affected micro-and macro-structures of pasta. Normal-and reduced-gluten pasta samples were characterized by similar water uptake during the first 10 min (25.6 ± 0.77 and 26.7 ± 0.64%, respectively) (see Appendix A). In contrast, differences in water uptake were found at the end of the test (65.2 ± 1.95 and 74.9 ± 1.42% for normal- and reduced-gluten pasta, respectively). The experimental optimal cooking time for normal- and reduced-gluten pasta were 6 and 5 min, respectively. At the end of the optimal cooking time, the highest absorption was found for reduced-gluten pasta (107 ± 0.3%) that showed a significantly higher value of cooking loss. The solid content in boiled water was 4.6 ± 0.2 and 5.4 ± 0.1% for normal- and reduced-gluten pasta, respectively.

### 3.5. Sensory Characteristics

The sensory properties of bread are shown in Figure 2A. Compared to the control, the use of the hydrolyzed flour was responsible for anincrease of the scores for cooked odor, fermented taste, and acid. The major differences between the two breads werethe sweetness and salty attributes. The values of cereal-like and yeast odor were similar between the two breads. Compared to bread with normalgluten content, the visual inspection of the bread with hydrolyzed flour showed a significant (*p* < 0.05) increase in crumb and crust color.

The sensory properties of pasta are shown in Figure 2B. Overall, the sensory profile of reduced-gluten pasta was described by fermented and corn-like odors and taste. Acid and salty profile differed significantly (*p* = 0.044) between the two types of pasta. The scores for stickiness and cooking homogeneity were slightly higher (*p* = 0.039) than those found for the normal-gluten pasta. 

### 3.6. In vivo Gluten Challenge

The CONSORT® 2010 flow diagram for randomized studies, representing the total number of people assessed for eligibility, enrolled, allocated into the two study arms after randomization and analysed, is shown in Figure 3. 

Of the 100 patients who participated in the first visit, 65 were excluded based on inclusion/exclusion criteria. Of the remaining 35 patients, 11 refused to participate and 24 entered the running-in phase without protocol deviations. At the final assessment, complete data were available for 20 of the 24 participants (83%), since two patients were non-compliant and two were lost to follow-up. All patients were tested for celiac disease (TTG and EMA) and wheat allergy (Skin prick test and/or wheat and gluten specific IgE). Native anti-gliadin antibodies (AGA-IgG and IgA) were available for 16 out of 20 patients and IgG were positive in 5 (31%) while IgA in 3 (18%). The recruitment and follow-up started in June 2014 and March 2015 and ended in June 2015 and March 2016, respectively.

Baseline demographic and clinical characteristics are shown in Appendix A

### 3.7. Clinical Scores

#### 3.7.1. Open Gluten-Free Diet

Compared to baseline, all patients after two weeks of GFD showed improvement of at least one of the IBS symptoms as estimated by IBS-SS, VAS, HDAS and IBS-QoL (Table 3). The IBS-SS and VAS were at baseline 237 ± 70.25 and 4.0 ± 1.54, respectively. Compared to baseline, IBS-SS and VAS significantly decreased after two weeks of run-in open GFD (237 ± 70.25 vs. 164.8 ± 79.15; *p* < 0.001, and 4.0 ± 1.54 vs. and 2.8 ± 1.40; *p* < 0.001 respectively). HDAS and IBS-QoL decreased from 22.9 ± 12.70 to 20.3 ± 10.65 (*p* < 0.028) and from 50.8 ± 27.94 to 39.5 ± 24.92 (*p* = 0.000) respectively.

#### 3.7.2. Double Blind Cross over challenge

As expected, the administration of normal-gluten bread and pasta increased (*p* = 0.000) IBS symptoms in all patients compared to the open GFD. Compared to normal-gluten, the administration of a reduced-gluten diet resulted in a decrease of the VAS score (*p* = 0.042), while no differences were found in the HDAS and IBS-QoL. Ten patients (50%) had and improvement of at least 30% of the VAS and/or IBS-SS and defined as responders (Appendix A). Principal Component Analysis based on all the IBS-SS, VAS, HDAS, and IBS-QoL data clearly differentiated responders and non-responders IBS patients (Appendix A). Responders were significantly younger (31.28 ± 1.5 vs 38.3 ± 2.5; *p* = 0.046) and less severe compared to non-responders. Permutation analysis showed the highest similarity between open the gluten-free and the reduced-gluten content diet versus thehabitual diet and normal-gluten content diet (Figure 4), irrespective of the responder’s status. No adverse events related to the study product were reported.

## 4. Discussion

In the present study, we describe a new protocol for the production of reduced content of gluten bread and pasta that have structural, image, color features and nutritional values similar to normal-gluten-content products; moreover, we demonstrate that its use in IBS patients is followed by a significant decrease of the disease severity scores and an improvement inquality of life.

During the last decade, the consumption of gluten containing foods has increased worldwide, especially in western countries [50]. Gluten, especially in high amounts (>15–20 g/die) [51] might be responsible of GI symptoms [12] such as abdominal pain and bloating, which are typical of IBS. On these bases it has been hypothesized that removing gluten from the diet might be beneficial for IBS patients. However, cereal based-foods are important sources of the daily intake of carbohydrates, proteins, dietary fibers, and vitamins, making it difficult to prescribe a GFD for nonceliac patients [28]. The reduction of gluten in the diet might be an option to consider. We are aware that gluten might not be the only cause of symptoms, and that FODMAP, ATI, and the nocebo effect may play a role in the pathogenesis of GI symptoms; nevertheless, we have focused our work on gluten, knowing that its reduction in the diet might be of help just for a subset of patients.

This study adapted the previously-established biotechnology [29,40,52] to partially degrade gluten (about 50%) without compromising the technological properties of the protein network. This level of gluten hydrolysis was achieved by the optimization of technological parameters such as dough yield, time of fermentation, and concentration of fungal proteases. The gluten reduction was obtained without any other intervention of wheat breeding or the endogenous proteolytic enzymes of flours [28]. After fermentation with sourdough and fungal proteases, the partially-hydrolyzed wheat flour with reduced-gluten was dried to obtain moisture contents similar to those before processing (about 14%). This drying process was in agreement with that routinely used in the bakery industries for manufacturing ready-to-use sourdough products [53]. This treated wheat matrix was used to manufacture bread and pasta. In contrast to naturally gluten-free flours or totally-hydrolyzed wheat flours [40,52,54], it does not need structuring agents (e.g., hydrocolloids) or other ingredients (e.g., starches and flavor enhancers) which negatively interfere with the sensory and nutritional features of the bread and pasta [55]. Moreover, we have shown no difference in the chemical and nutritional characteristics of reduced—as compared to the normal—gluten content bread and pasta. All the values that characterized the bread and pasta with reduced gluten contents were in the range typical for traditional wheat with normal levelsof gluten; however, due to the gluten hydrolysis, the specific volume, structural properties, and some sensory features of bread were lower for reduced-gluten content bread [56]. Overall, the texture features of cooked pasta are key attributes for itsassessment [57]. Consumers prefer pasta that is firm, chewy, and not gummy or adhesive. After cooking, hardness was highest in pasta made with normal levels of gluten, supporting the key role of non-hydrolyzed gluten. All the other texture properties (cohesiveness, springiness, resilience, gumminess, and chewiness) were similar to those of pasta made with normal levels of gluten. The ability of pasta to absorb water is one of the most important characteristics, and depends on raw materials and technological parameters which may increase the formation of micro- and macrostructures that are related to porosity and structural features. Pasta made with reduced gluten levels showed fast and relevant absorption of water as compared to pasta with normal level of gluten; this is possibly explained by the increased amount of hydrophilic molecules (such as free amino acids and small peptides) produced during gluten proteolysis [40], rather than bythe effect of the technological parameters (e.g., drying conditions) [58]. According to the sensory analysis, pasta made with reduced gluten was similar to the pasta made with normal levels of gluten, with the exception of theacidity and salty profiles, which were the highest in the former. Probably, these differences are secondary to the highest amount of organic acids and free amino acids in pasta with reduced gluten contents. The observed changes in taste, texture, and appearance of reduced gluten products is one limitation of this study; since all patients were exposed to both products, the blinding of this study might have been lost [24]. These attributes did not affect the overall acceptance of RG pasta. 

It has been hypothesized that some patients with IBS may benefit from GFD despite not having celiac disease and/or from the reduction of FODMAPs in the diet. However, systematic reviews of studies evaluating a GFD and a low FODMAP diet have provided conflictingconclusions [20,22,23], suggesting that more data are needed. Although a strict GFD in the absence of celiac disease should not be started, a reduction of gluten content might be a valid option; therefore, in view of our documented experience in the field of hydrolysis of gluten peptides [29,40,52,54], in the present paper, we decided to test whether the use of a reduced-gluten content diet might be of help in symptoms improvement in patients with IBS. We were able to show that after two weeks of a reduced-gluten diet, patients with IBS showed an improvement of the severity score (VAS) that determined a treatment success in only half of the patients. The observation that the improvement was significantly higher while patients were following an open GFD underlines the strong nocebo effect operating in this type of dietetic intervention study. 

Recent scientific evidences suggest that other wheat components such as FODMAPs, ATIs, and wheat germ agglutinins might trigger the symptoms of IBS patients [14,15,16,17,18]. There are several ways in which food components may trigger symptoms: it has been shown that in the gut lumen, the interaction between dietary factors may be the optima substrate for microbiota digestion, resulting in an increase of water volume and colonic gas production, and/or triggering the release of inflammatory mediators and the activation of the immune system that, in turn, can stimulate mechanoreceptors and sensory nerve pathways responsible of visceral hypersensitivity and the generation of functional GI symptoms [59]. 

Under our experimental conditions, FODMAPs were not hydrolyzed during fermentation of doughs mainly because of specific biotechnology that we have selected for gluten hydrolyses [29,40,52] and the short time of fermentation while we have reached a partial hydrolysis of ATIs. Based on these results, we may speculate that gluten and ATIs play a role in symptom aggravation in IBS patients, while no conclusion maybe drawn for FODMAPs. We are aware that under the umbrella of IBS, there may be patients who are intolerant to gluten, α-amylase/trypsin inhibitors, FODMAPs or other triggers causing innate and/or adaptive immune and non-immune responses responsible for gastrointestinal symptoms [60], and this might explain the incomplete response to our dietetic intervention. Previous data have shown that specific sourdough systems or lengthened yeast fermentation causes not only gluten to cleave in grain products, but also induces a reduction of FODMAPs [24,61]; this is reflected in the alleviation of symptoms. It was shown that sourdough wheat bread with lower amounts of FODMAPs and ATIs was not effective to reduce the GI symptoms and markers of inflammation in IBS patients compared to yeast wheat bread [24]. In contrast, other studies reported that a reduction of FODMAPs from the diet was correlated with improved symptoms and quality of life of IBS patients [1,62]. In a double-blind, randomized, placebo-controlled, crossover trial, the addition of a GFD to IBS patients already on a low FODMAP diet did not improve IBS symptoms [63]. 

Interestingly, besides causing CD and wheat allergy, gluten, or more broadly wheat, has been advocated as the possible cause of a new condition known as non-celiac gluten (or wheat) sensitivity, that is characterized by a combination of GI and extra-GI symptoms that disappear after gluten withdrawal in patients in whom both CD and wheat allergy have been correctly excluded [48].This new entity can be the next future candidate for dietetic treatment with RGC products, since the threshold has not yet been established of gluten’srole in causing thesymptoms in these patients who,theoretically, the do not requirea GFD which is as strict as that for celiac patients.

The strengths of present study are the following. This is the first randomized, double-blind, cross-over trial to use wheat-based products with intermediate contents of gluten in patients with IBS supporting a reduced intake of gluten as a new dietetic strategy to reduce symptoms and improve Quality of Life in a subset of IBS patients. Secondly, a gluten-free diet in the absence of celiac disease might be considered unethical for several reasons (inability to correctly diagnose celiac disease, nutritional imbalances and possibly deficiencies, high economic burden related to an unjustified gluten free diet, and social withdrawal) and should be discouraged; however, the possibility of achieving a clinical benefit following a reduced-gluten content diet without the drawbacks of the gluten free diet is fascinating. Finally, the absence of structuring agents or other ingredients used to produce gluten free products increases the healthy profile of reduced-gluten content bread and pasta produced according to our protocol.

However, there are several limitations that need to be acknowledged. Our study shall be considered as an exploratory trial, since the small sample size and the limited time of the intervention do not allow us to draw definitive conclusions,and larger trials are needed before suggesting this new dietetic approach. Moreover, we are not able to discriminate which IBS-subtype might the best candidate for the reduced-gluten content diet, although younger patients with less severe symptoms seem to be the best candidates. Finally, the major limit is represented by the limited choice of products that we were able to offer to patients,whichmight have biased the final result.

Data herein reported are novel encouraging findings that should spur a new avenue of research aimed atsetting up new protocols for the production of reducedgluten, ATIs, and FODMAPs products to be tested in future challenge studies that could be of benefit for patients with IBS and non-celiac gluten sensitivity, considering the possible implication that non-digestible carbohydrates present in the foods we eat,as well aspolyphenols with modest bioavailability, reach the colon unaltered, and exert potential effects on the gut microbiota [4,64], with consequences that still remain to explore. 

## Figures and Tables

**Figure 1 nutrients-10-01873-f001:**
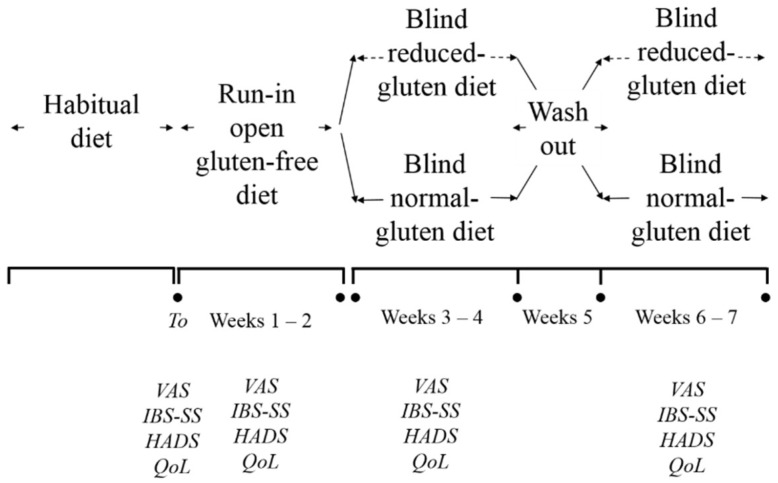
Crossover design of the study and the timing of clinical evaluations. IBS-SS, Irritable Bowel Syndrome Severity Score; VAS, Visual Analogue Scale; HADS, Hospital Anxiety and Depression Scale; QoL, Quality of Life.

**Figure 2 nutrients-10-01873-f002:**
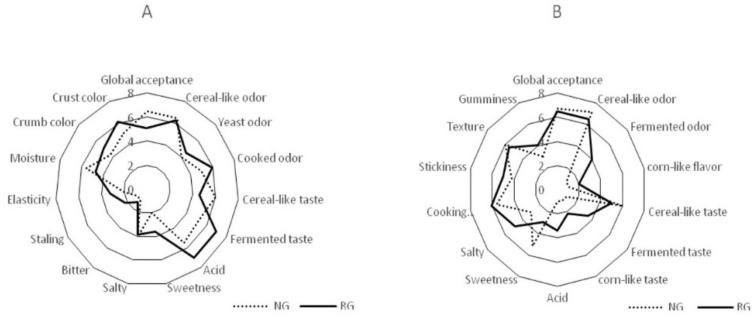
Sensory analysis of bread (**A**) and pasta (**B**) made with normal (NG) or reduced content of gluten (RG) flours. Reduced-gluten wheat flour was fermented with fungal proteases and selected lactobacilli at 30 °C for 8 h. Data are the means of three independent analyses.

**Figure 3 nutrients-10-01873-f003:**
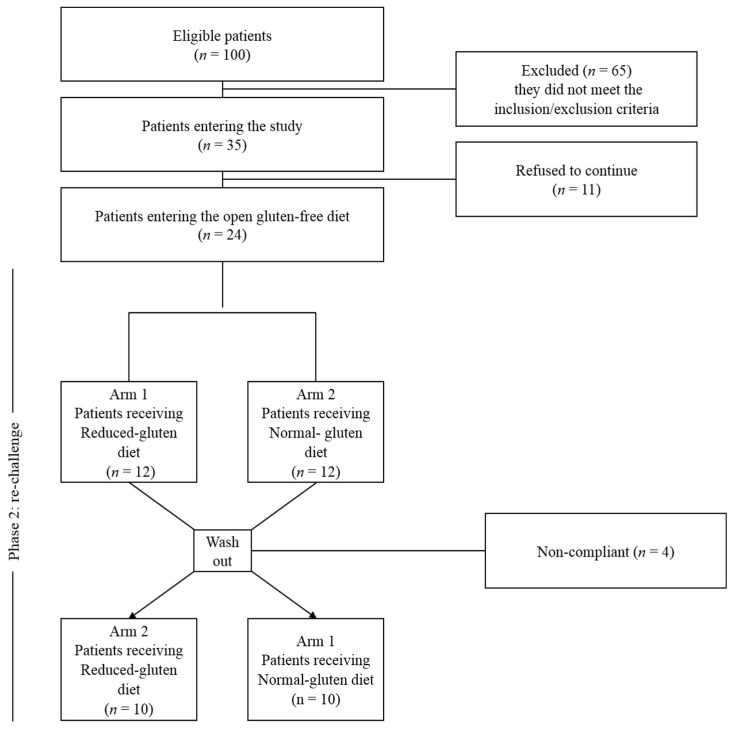
Flow diagram of patients into the trial from eligibility to the end of gluten challenge.

**Figure 4 nutrients-10-01873-f004:**
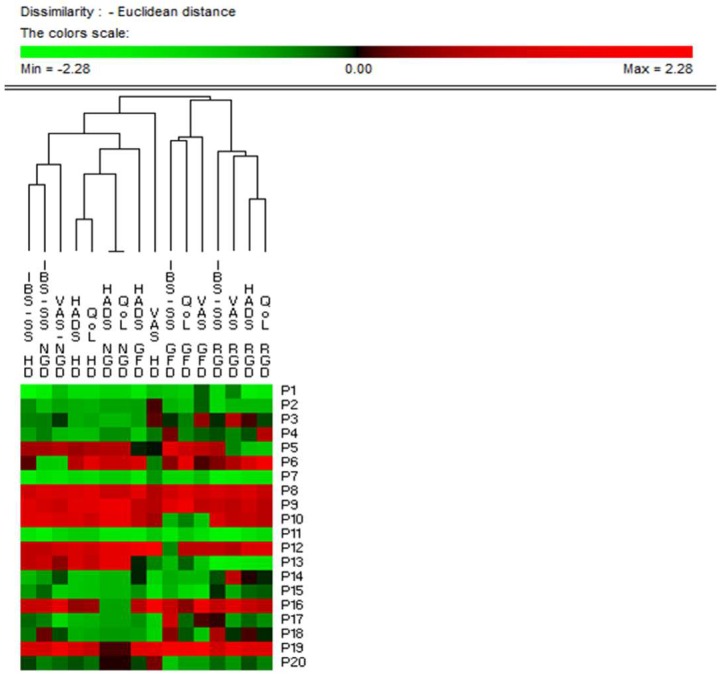
Permutation analysis based on clinical scores. Euclidean distance and McQuitty’s criterion were used for clustering. Cases (patients) were numbered from P1 to P20. Variables were reported as abbreviation of clinical scores (IBS-SS, Irritable Bowel Syndrome Severity Score; VAS, Visual Analogue Scale; HADS, Hospital Anxiety and Depression Scale; QoL, Quality of Life) plus the type of diet (HD, Habitual Diet; GFD, Run-in Open gluten-free diet; RGD, Reduced-gluten diet; NGD, Normal-gluten diet). Colours correspond to normalized mean data levels from low (green) to high (red).

**Table 1 nutrients-10-01873-t001:** Chemical composition and nutritional indexes of bread and pasta made with normal-(NG) and reduced-(RG) gluten content flours. RG wheat flour was fermented with fungal proteases and selected lactobacilli at 30 °C for 8 h.

	NG Bread ^1^	RG Bread	*p* Value NG vs. RG Bread	NG Pasta	RG Pasta	*p* Value NG vs. RG Pasta
Energetic value (Kcal/100 g)	258 ± 5.67	262 ± 2.88	*p* = 0.112	360 ± 15.48	367 ± 8.07	*p* = 0.204
Moisture (%)	29.0 ± 0.2	29.9 ± 0.3	*p* = 0.104	10.2 ± 0.4B	10.5 ± 0.8	*p* = 0.361
Fat (%)	1.23 ± 0.01	1.21 ± 0.05	*p* = 0.288	0.44 ± 0.01	0.45 ± 0.02	*p* = 0.098
Fatty acid methyl esters (% of fat)						
C16:0	17.1 ± 0.3	18.3 ± 0.26	*p* = 0.187	16.7 ± 0.22	17.3 ± 0.13	*p* = 0.305
C16:1	0.6 ± 0.03	1 ± 0.05	*p* = 0.081	0.5± 0.02	0.4± 0.03	*p* = 0.355
C18:0	2.2 ± 0.08	2.3 ± 0.03	*p* = 0.222	2.0± 0.04	2.1 ± 0.04	*p* = 0.155
C18:1	37.1 ± 1.64	37.2 ± 1.12	*p* = 0.093	35.0 ± 0.84	33.3 ± 0.49	*p* = 0.111
C18:2	40.4 ± 1.73	38.9 ± 0.24	*p* = 0.174	41.9 ± 1.64	42.5 ± 1.91	*p* = 0.161
Saturated fatty acids (% of DM)	19.2 ± 0.34	20.6 ± 0.37	*p* = 0.442	19.6 ± 0.84	19.4 ± 0.21	*p* = 0.313
Monounsaturated fatty acids (% of DM)	38.3 ± 1.37	38.2 ± 1.64	*p* = 0.177	35.7 ± 0.69	33.5 ± 1.13	*p* = 0.657
Polyunsaturated fatty acids (% of DM)	42.5 ± 0.83	41.2 ± 0.91	*p* = 0.405	44.7 ± 1.61	49.1 ± 2.11	*p* = 0.318
Total carbohydrate (% of DM)	55.11 ± 0.32	53.54 ± 0.96	*p* = 0.222	76.11 ± 0.83	81.4 ± 1.79	*p* = 0.189
FODMAPs (g/100 g)	0.96 ± 0.015	0.90 ± 0.018	*p* = 0.182	0.74 ± 0.007	0.70 ± 0.005	*p* = 0.099
Protein (N x 6.25) (g/100 g)	7.98 ± 0.34	7.76 ± 0.17	*p* = 0.242	11.32 ± 0.48	11.64 ± 0.50	*p* = 0.163
Crude fiber content (% of DM)	2.2 ± 0.02	2.86 ± 0.12	*p* = 0.114	3.17 ± 0.03	3.54 ± 0.15	*p* = 0.106
Na (% of DM)	0.85 ± 0.07	0.81 ± 0.03	*p* = 0.204	0.43 ± 0.06	0.45± 0.05	*p* = 0.101
Cl (% of DM)	0.87 ± 0.00	0.77 ± 0.00	*p* = 0.078	0.46 ± 0.01	0.44 ± 0.00	*p* = 0.085
K (mg/Kg)	1216 ± 23.83	1171 ± 50.35	*p* = 0.063	2953 ± 64.96	3003 ± 66.06	*p* = 0.072
Ca (mg/Kg)	364 ± 15.65	354 ± 12.74	*p* = 0.065	605 ± 6.65	401 ± 9.62	*p* = 0.042
Mg (mg/Kg)	142 ± 6.11	138 ± 2.71	*p* = 0.088	422 ± 9.28	480 ± 20.64	*p* = 0.074
P (mg/Kg)	982 ± 10.81	908 ± 21.79	*p* = 0.219	2615 ± 51.25	2670 ± 48.06	*p* = 0.252
Fe (mg/Kg)	5.3 ± 0.21	5.1 ± 0.11	*p* = 0.547	11.4 ± 0.12	10.1 ± 0.36	*p* = 0.463

^1^ Data are the mean of three independent fermentations twice analyzed.

**Table 2 nutrients-10-01873-t002:** Structural and image and color characteristics of bread and pasta made with normal-(NG) and reduced-(RG) gluten content flours. RG wheat flour was fermented with fungal proteases and selected lactobacilli at 30 °C for 8 h.

	NG Bread ^1^	RG Bread	*p* Value
	Structural Characteristics		
*Hardness* (g)	4970 ± 256	5304 ± 204	*p =* 0.291
*Resilience*	1.01 ± 0.01	0.91 ± 0.03	*p* = 0.014
*Fracturability* (g)	3526 ± 166	2940 ± 87	*p* = 0.022
Specific volume (cm^3^/g)	2.3 ± 0.01	2.2 ± 0.02	*p* = *0.047*
	Image Analysis		
Black pixel area (%)	40.3 ± 0.2	38.9 ± 0.3	*p* = 0.031
	Color Analysis		
*L*	58 ± 2.08	42 ± 4.16	*p* = 0.043
a	7.29 ± 0.52	12 ± 1.18	*p* = 0.022
b	29.45 ± 0.18	28.88±0.25	*p* = 0.111
d*E*	44.30 ± 1.72	58.58 ± 1.76	*p* = 0.037
	**NG Pasta**	**RG Pasta**	
	Structural Characteristics		
*Hardness* (N)	5.10 ± 0.05	4.89 ± 0.02	*p* = 0.034
*Cohesiveness*	0.81 ± 0.02	0.75 ± 0.01	*p* = 0.094
*Springiness*	0.62 ± 0.01	0.60 ± 0.01	*p* = 0.103
*Resilience*	0.55 ± 0.01	0.52 ± 0.01	*p* = 0.157
*Gumminess* (N)	4.13 ± 0.04	4.11 ± 0.07	*p* = 0.146
*Chewiness* (N)	2.56 ± 0.03	2.45 ± 0.04	*p* = 0.254
	Color Analysis		
*L*	65 ± 1.01	59 ± 1.15	*p* = 0.026
a	6.92 ± 0.08	6.45 ± 0.12	*p* = 0.013
b	35.8 ± 0.14	34.9 ± 0.23	*p* = 0.011
d*E*	43.22 ± 1.38	46.71 ± 1.48	*p* = 0.008

^1^ Data are the mean of three independent fermentations twice analyzed.

**Table 3 nutrients-10-01873-t003:** Modification of the clinical scores^1^ in the IBS patients in the different diets according to the challenge results.

Clinical Scores	Diet	Average Value ± SD	*p* Value				
				Habitual Diet vs. Run-in Open Gluten-Free Diet	Run-in Open Gluten-Free Diet vs. Reduced-Gluten Diet	Run-in Open-Gluten-Free diet vs. Normal-Gluten Diet	Reduced-Gluten Diet vs. Normal-Gluten Diet
IBS-SS	Habitual diet	237.3 ± 70.25	IBS-SS	0.000	0.002	0.000	0.166
	Run-in Open gluten-free diet	164.8 ± 79.15					
	Reduced-gluten diet	214.4 ± 97.59					
	Normal-gluten diet	229.5 ± 79.82					
VAS	Habitual diet	4.0 ± 1.54	VAS	0.000	0.000	0.000	0.042
	Run-in Open gluten-free diet	2.8 ± 1.40					
	Reduced-gluten diet	3.7 ± 1.77					
	Normal-gluten diet	4.3 ± 1.67					
HADS	Habitual diet	22.9 ± 12.70	HADS	0.027	0.314	0.423	0.357
	Run-in Open gluten-free diet	20.3 ± 10.65					
	Reduced-gluten diet	19.7 ± 10.82					
	Normal-gluten diet	20.7 ± 12.35					
QoL	Habitual diet	50.8 ± 27.94	QoL	0.000	0.000	0.046	0.357
	Run-in Open gluten-free diet	39.5 ± 24.92					
	Reduced-gluten diet	45.2 ± 26.43					
	Normal-gluten diet	47.3 ± 24.69					

^1^ IBS-SS, Irritable Bowel Syndrome Severity Score; VAS, Visual Analogue Scale; HADS, Hospital Anxiety and Depression Scale; QoL, Quality of Life.

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
