# Peer review of "New Protocol for Production of Reduced-Gluten Wheat Bread and Pasta and Clinical Effect in Patients with Irritable Bowel Syndrome: A randomised, Double-Blind, Cross-Over Study"

_nutrients, 2018, doi:10.3390/nu10121873_

Reviewer 1 Report

A generally interesting and informative. However, certain aspects of the manuscript are underdeveloped, and some concerns and issues have to be resolved.

 Specific comments:

- Please change "Randomized Double Blind Cross Over Study" to "A randomised, double-blind, cross-over study".

- Please spell the abbreviation FODMAP out in full in the abstract.

- Please change "responsible of” to "responsible for".

- Psychosocial stressors likely play an important role in irritable bowel syndrome, with a recent meta-analysis reporting an association between IBS and posttraumatic stress disorder (citation: Ng QX, Soh AY, Loke W, Venkatanarayanan N, Lim DY, Yeo WS. Systematic review with metaanalysis: The association between posttraumatic stress disorder and irritable bowel syndrome. Journal of gastroenterology and hepatology. 2018 Aug 25).

- Please change "may result in a clinical benefit" to "may result in clinical benefit".

- Please change “Recently, other systematic review” to “Recently, a systematic review”.

- How was sample size determined? There is currently no evidence of power calculation.

- The current exclusion criteria is too broad and generic. What does it mean to have “clinically significant cardiovascular, hepatic, respiratory, endocrine, renal, hematologic, neurologic or psychiatric disorder”? Please be more specific.

- Please provide exact P values. Reporting P values simply as P < .05 (or any other threshold) is not accurate or informative.

- Please change “Despite causing” to “In addition to causing”.

- Please change “several limits” to “several limitations”.

- Please change “do not allow to drawn definitive conclusions” to “do not allow us to draw definitive conclusions.”

- Did authors test for success of blinding?

- Were any of the patients tested for gluten sensitivity? There's some evidence that the AGA-IgA and the AGG-IgG blood tests could indicate non-celiac gluten sensitivity (citation: Volta U, Tovoli F, Cicola R, Parisi C, Fabbri A, Piscaglia M, Fiorini E, Caio G. Serological tests in gluten sensitivity (nonceliac gluten intolerance). Journal of clinical gastroenterology. 2012 Sep 1;46(8):680-5). Gluten intolerance is fairly common and can cause widespread symptoms. Based on the different dietary factors responsible for symptom generation, patients can be labelled non-coeliac gluten sensitive or more broadly non-coeliac wheat or wheat protein sensitivity or, even, FODMAP sensitive.

- Until recently gluten intolerance has been believed to be typical of celiac disease and wheat allergy. In the last few years, however, several study results have been published that have proved that gluten intolerance can also affect people who do not suffer from any of the above mentioned diseases (citation: Catassi C, Elli L, Bonaz B, Bouma G, Carroccio A, Castillejo G, Cellier C, Cristofori F, de Magistris L, Dolinsek J, Dieterich W. Diagnosis of non-celiac gluten sensitivity (NCGS): the Salerno experts’ criteria. Nutrients. 2015 Jun 18;7(6):4966-77). The new syndrome has been named non-celiac gluten sensitivity or gluten sensitivity. This should be mentioned in the discussion.

- In the discussion, some key pathophysiological mechanisms should be discussed. For example, in the gut lumen, the interaction between dietary factors (carbohydrates, lipids and proteins) and the microbiota results in gas production and/or passage of noxious macromolecules triggering the release of mast cell mediators and the activation of the immune system. These mechanisms are at the base of mechanoreceptor and sensory nerve pathway activation ultimately responsible for commonly reported symptoms, such as abdominal pain, bloating and distension, especially in genetically predisposed patients.

- For future work, other food products that could benefit IBS sufferers should be investigated as well. For example, non-digestible carbohydrates (prebiotics) that are in the foods we eat (citation: Singh RK, Chang HW, Yan D, Lee KM, Ucmak D, Wong K, Abrouk M, Farahnik B, Nakamura M, Zhu TH, Bhutani T. Influence of diet on the gut microbiome and implications for human health. Journal of translational medicine. 2017 Dec;15(1):73) and polyphenols (most polyphenols have little bioavailability and reach the colon almost unaltered, exerting potential effects on the gut microbiota, citation: Ng QX, Soh AYS, Loke W, Venkatanarayanan N, Lim DY, Yeo WS. A Meta-Analysis of the Clinical Use of Curcumin for Irritable Bowel Syndrome (IBS). Journal of clinical medicine. 2018 Sep 22;7(10):298). There is a recent meta-analysis, which reported potentially beneficial effects of curcumin on IBS symptoms.

 Author Response

Response to Reviewer 1 Comments

Point 1. A generally interesting and informative. However, certain aspects of the manuscript are underdeveloped, and some concerns and issues have to be resolved.

Response 1. First of all, I would like to thank you for having contributed to the improvement of the manuscript. All modifications required were made in the revised manuscript.

Point 2. Please change "Randomized Double Blind Cross Over Study" to "A randomised, double-blind, cross-over study".

Response 2. - OK, the title was modified (see p. 1, l.4-5)

Point 3. Please spell the abbreviation FODMAP out in full in the abstract.

Response 3. - OK, “Fermentable oligo-, di- and monosaccharides and polyols”was used in the abstract instead of “FODMAP”(see p. 1, l.22-23).

Point 4. Please change "responsible of” to "responsible for".

Response 4. OK, "responsible for" was used instead of"responsible of”  (see p. 1, l.41-42).

Point 5. Psychosocial stressors likely play an important role in irritable bowel syndrome, with a recent meta-analysis reporting an association between IBS and posttraumatic stress disorder (citation: Ng QX, Soh AY, Loke W, Venkatanarayanan N, Lim DY, Yeo WS. Systematic review with metaanalysis: The association between posttraumatic stress disorder and irritable bowel syndrome. Journal of gastroenterology and hepatology. 2018 Aug 25).

Response 5. OK, a new sentence and the relative citation were included in the revised manuscript  (see p. 1 l.41-43).

Point 6. Please change "may result in a clinical benefit" to "may result in clinical benefit".

Response 6. OK, "may result in clinical benefit " was used instead of "may result in a clinical benefit”(see p. 2  l.54).

Point 7. Please change “Recently, other systematic review” to “Recently, a systematic review”.

Response 7. OK, “Recently, a systematic review” was used instead of “Recently, other systematic review” (see p. 2 l.63).

Point 8. How was sample size determined? There is currently no evidence of power calculation.

Response 8. OK, more details were added regarding the sample size and the power calculation (see p. 7 l. 286-288).

Point 9. The current exclusion criteria is too broad and generic. What does it mean to have “clinically significant cardiovascular, hepatic, respiratory, endocrine, renal, hematologic, neurologic or psychiatric disorder”? Please be more specific.

Response 9. I am sorry for the mistake; the text was revised (see p. 5 l. 229-235).

Point 10. Please provide exact P values. Reporting P values simply as P < .05 (or any other threshold) is not accurate or informative.

Response 10. OK, the exact P values were added in the revised manuscript including tables (see p. 9 l.339, 340; p.10 l.353, 354,364; Table 1, Table 2).

Point 11. Please change “Despite causing” to “In addition to causing”.

Response 11. OK, the sentence was modified using “In addition to causing” (see p. 15 l. 450).

Point 12. Please change “several limits” to “several limitations”.

Response 12. OK, “several limits” was changed to “several limitations” (see p. 17 l. 548).

Point 13. Please change “do not allow to drawn definitive conclusions” to “do not allow us to draw definitive conclusions.”

Response 13. OK, “do not allow to drawn definitive conclusions” was changed to “do not allow us to draw definitive conclusions.” (see p. 17 l. 550).

Point 14. Did authors test for success of blinding?

Response 14. OK, The success of blinding has been verified during two different panel test performed before the study. Despite some differences in sensory characteristics (see paragraph 3.5) 10 out of 10 participants at the panel test were unable to distinguish which bread and pasta was the low gluten content one(see p. 6 l.253-255).

Point 15. Were any of the patients tested for gluten sensitivity? There's some evidence that the AGA-IgA and the AGG-IgG blood tests could indicate non-celiac gluten sensitivity (citation: Volta U, Tovoli F, Cicola R, Parisi C, Fabbri A, Piscaglia M, Fiorini E, Caio G. Serological tests in gluten sensitivity (nonceliac gluten intolerance). Journal of clinical gastroenterology. 2012 Sep 1;46(8):680-5). Gluten intolerance is fairly common and can cause widespread symptoms. Based on the different dietary factors responsible for symptom generation, patients can be labelled non-coeliac gluten sensitive or more broadly non-coeliac wheat or wheat protein sensitivity or, even, FODMAP sensitive.

Response 15. OK, The following sentence has been added in the result section: “All the patients were tested for celiac disease (TTG and EMA) and for wheat allergy (Skin prick test and/or wheat and gluten specific IgE). Native anti-gliadin antibodies (AGA-IgG and IgA) were available for 16 out of 20 patients and IgG were positive in 5 (31%) while IgA in 3 (18%). We didn’t find any correlation between the AGA positivity and the clinical response”(see p. 12 l. 404-406).

Point 16. Until recently gluten intolerance has been believed to be typical of celiac disease and wheat allergy. In the last few years, however, several study results have been published that have proved that gluten intolerance can also affect people who do not suffer from any of the above mentioned diseases (citation: Catassi C, Elli L, Bonaz B, Bouma G, Carroccio A, Castillejo G, Cellier C, Cristofori F, de Magistris L, Dolinsek J, Dieterich W. Diagnosis of non-celiac gluten sensitivity (NCGS): the Salerno experts’ criteria. Nutrients. 2015 Jun 18;7(6):4966-77). The new syndrome has been named non-celiac gluten sensitivity or gluten sensitivity. This should be mentioned in the discussion.

Response 16. OK, The following sentence has been added in the discussion: “Interestingly, beside causing CD and wheat allergy, gluten or more broadly wheat has been advocate the possible cause of an new condition known as non-celiac gluten (or wheat) sensitivity that is characterized by a combination of GI and extra-GI symptoms that disappear after gluten withdrawal in patients in whom both CD and wheat allergy have been correctly excluded (Catassi C, Elli L, Bonaz B, Bouma G, Carroccio A, Castillejo G, Cellier C, Cristofori F, de Magistris L, Dolinsek J, Dieterich W. Diagnosis of non-celiac gluten sensitivity (NCGS): the Salerno experts’ criteria. Nutrients. 2015;7:4966). This new entity can be the next future candidate to a dietetic treatment with RGC products since it has not been yet established the threshold of gluten responsible of symptoms in these patients and theoretically they do not need a GFD as strict as celiac patients”(see p. 16 l.530-536).

Point 17. In the discussion, some key pathophysiological mechanisms should be discussed. For example, in the gut lumen, the interaction between dietary factors (carbohydrates, lipids and proteins) and the microbiota results in gas production and/or passage of noxious macromolecules triggering the release of mast cell mediators and the activation of the immune system. These mechanisms are at the base of mechanoreceptor and sensory nerve pathway activation ultimately responsible for commonly reported symptoms, such as abdominal pain, bloating and distension, especially in genetically predisposed patients.

Response 17. OK, The following sentence has been added in the discussion: “There are several way in which food components may trigger symptoms: it has been shown that in the gut lumen, the interaction between dietary factors may be the optima substrate for microbiota digestion resulting in an increase of water volume and colonic gas production and/or triggering the release of inflammatory mediators and the activation of the immune system that in turn can stimulate mechanoreceptor and sensory nerve pathway responsible of visceral hypersensitivity and the generation of functional GI symptoms (Staudacher, H. M., & Whelan, K. The low FODMAP diet: recent advances in understanding its mechanisms and efficacy in IBS. Gut. 2017;66:1517) (see p. 16 l. 507-512).

Point 18. For future work, other food products that could benefit IBS sufferers should be investigated as well. For example, non-digestible carbohydrates (prebiotics) that are in the foods we eat (citation: Singh RK, Chang HW, Yan D, Lee KM, Ucmak D, Wong K, Abrouk M, Farahnik B, Nakamura M, Zhu TH, Bhutani T. Influence of diet on the gut microbiome and implications for human health. Journal of translational medicine. 2017 Dec;15(1):73) and polyphenols (most polyphenols have little bioavailability and reach the colon almost unaltered, exerting potential effects on the gut microbiota, citation: Ng QX, Soh AYS, Loke W, Venkatanarayanan N, Lim DY, Yeo WS. A Meta-Analysis of the Clinical Use of Curcumin for Irritable Bowel Syndrome (IBS). Journal of clinical medicine. 2018 Sep 22;7(10):298). There is a recent meta-analysis, which reported potentially beneficial effects of curcumin on IBS symptoms.

Response 18. OK, We modified the conclusive sentence taking on to account the suggestion of the reviewer: “Data herein reported are novel encouraging findings that should spur a new avenue of research aimed to set-up of new protocols for the production of reduced-gluten, ATIs and FODMAPs products to be tested in future challenge studies that could be of benefit for patients with IBS and non-celiac gluten sensitivity considering the possible implication that non-digestible carbohydrates present in the foods we eat and polyphenols with modest bioavailability reach the colon unaltered and exert potential effects on the gut microbiota (Singh RK, Chang HW, Yan D, Lee KM, Ucmak D, Wong K, Abrouk M, Farahnik B, Nakamura M, Zhu TH, Bhutani T. Influence of diet on the gut microbiome and implications for human health. Journal of translational medicine. 2017;15:73) with consequences that still remain to explore” (see p.17 l. 555-560). 

Reviewer 2 Report

The manuscript contains some interesting findings. However, mechanistic points of the effects of reduced-gluten, wheat bread and pasta on the symptoms of IBS patients should be briefly mentioned in the Discussion section.

Author Response

Response to Reviewer 2 Comments

 Point. The manuscript contains some interesting findings. However, mechanistic points of the effects of reduced-gluten, wheat bread and pasta on the symptoms of IBS patients should be briefly mentioned in the Discussion section.

Response. First of all, I would like to thank you for having contributed to the improvement of the manuscript.

The manuscript was checked by a native English speaking colleague.

The discussion was revised by adding the mechanistic points of the effects of reduced-gluten, wheat bread and pasta on the symptoms of IBS patients (see p. 16 l. 505-512).
